# Conditioned Medium of Mesenchymal Stromal Cells Loaded with Paclitaxel Is Effective in Preclinical Models of Triple-Negative Breast Cancer (TNBC)

**DOI:** 10.3390/ijms24065864

**Published:** 2023-03-20

**Authors:** Nicoletta Cordani, Daniela Lisini, Valentina Coccè, Giuseppe Paglia, Ramona Meanti, Maria Grazia Cerrito, Pietro Tettamanti, Luca Bonaffini, Francesca Paino, Giulio Alessandri, Angela Marcianti, Aldo Giannì, Chiara Villa, Mario Mauri, Luca Mologni, Antonio Torsello, Augusto Pessina, Marina Elena Cazzaniga

**Affiliations:** 1School of Medicine and Surgery, Milano-Bicocca University, 20900 Monza, Italy; 2Cell Therapy Production Unit-UPTC, Fondazione IRCCS Istituto Neurologico Carlo Besta, 20133 Milan, Italy; 3CRC StaMeTec, Department of Biomedical, Surgical and Dental Sciences, University of Milan, 20122 Milan, Italy; 4Department of Molecular Medicine, University of Pavia, 27100 Pavia, Italy; 5Maxillo-Facial and Dental Unit, Fondazione Ca’ Granda IRCCS Ospedale Maggiore Policlinico, 20122 Milan, Italy; 6Phase 1 Research Centre, Fondazione IRCCS San Gerardo dei Tintori, Via Pergolesi 33, 20900 Monza, Italy

**Keywords:** paclitaxel, mesenchymal stromal cells, triple-negative breast cancer

## Abstract

Triple-negative breast cancer (TNBC) is a very aggressive disease even in its early stages and is characterized by a severe prognosis. Neoadjuvant chemotherapy is one of the milestones of treatment, and paclitaxel (PTX) is among the most active drugs used in this setting. However, despite its efficacy, peripheral neuropathy occurs in approximately 20–25% of cases and represents the dose-limiting toxicity of this drug. New deliverable strategies to ameliorate drug delivery and reduce side effects are keenly awaited to improve patients’ outcomes. Mesenchymal stromal cells (MSCs) have recently been demonstrated as promising drug delivery vectors for cancer treatment. The aim of the present preclinical study is to explore the possibility of a cell therapy approach based on the use of MSCs loaded with PTX to treat TNBC-affected patients. For this purpose, we in vitro evaluated the viability, migration and colony formation of two TNBC cell lines, namely, MDA-MB-231 and BT549, treated with MSC-PTX conditioned medium (MSC-CM PTX) in comparison with both CM of MSCs not loaded with PTX (CTRL) and free PTX. We observed stronger inhibitory effects on survival, migration and tumorigenicity for MSC-CM PTX than for CTRL and free PTX in TNBC cell lines. Further studies will provide more information about activity and potentially open the possibility of using this new drug delivery vector in the context of a clinical study.

## 1. Introduction

Triple-negative breast cancer (TNBC) is a subtype of breast cancer defined by the lack of expression of Estrogen-Receptor (ER), Progesterone-Receptor (PgR) and Human-Epidermal Growth Factor Receptor-2 (HER2) [1]. It is characterized, even in its early stages, by a very aggressive behavior, with a peak in relapses within the first 3 years from diagnosis. Metastatic progression is typically marked by early relapse and the predominance of hepatic, pulmonary and central nervous system metastasis [2].

Neoadjuvant chemotherapy has become, over the years, the standard of care for TNBC patients. Different trials [3,4,5] have demonstrated that the most active drugs in this setting are anthracyclines, taxanes and platinum compounds, leading to pathological complete response (pCR) rates of approximately 35–40%. More recently, the addition of Pembrolizumab to a combination of carboplatin and paclitaxel (PTX) followed by epirubicin-cyclophosphamide (EC) was demonstrated to improve pCR rates and event-free survival (EFS) [6]. It is clear that taxanes still continue to be important players in the game of improving the survival of early-stage TNBC patients. However, taxane-induced peripheral neuropathy (TIPN), characterized by sensory and motor neuropathy symptoms, is a frequent consequence linked to taxane-based chemotherapy [7]. So far, the improvement of the taxane toxicity profile, by binding the drug with albumin, has been an important step forward; however, new technologies could further help to improve this severe toxicity.

Recently, mesenchymal stromal cells (MSCs) have been demonstrated to be promising drug delivery vectors for the treatment of cancer and other diseases [8,9,10,11]. MSCs can be isolated from several mammalian tissues (e.g., adipose tissue, bone marrow, skin, umbilical cord blood, etc.). More importantly, MSCs are able to home to inflammatory microenvironments and migrate to tumor masses after systemic injection [12,13,14]. We have recently shown that MSCs are capable of delivering drugs without genetic manipulation [13]. We found that MSCs derived from different tissues (bone marrow, adipose tissue and gingival papilla) upon in vitro incubation with PTX incorporated significant amounts of the drug that were subsequently released into the culture medium, also through exosome release [15,16,17,18].

Therefore, MSCs can acquire strong anti-tumor activity through their capacity to take up PTX and release it at the proper site.

The use of MSCs to transport and deliver anti-cancer drugs should have several potential advantages over the use of free-form drugs or other delivery systems: (1) This approach may increase protection of the drug from degradation before reaching the target cancer cells. (2) Given that MSCs are capable of integrating into tumor stroma (MSC homing capacity), they may enhance drug concentration in the tumor environment and thus increase tumor drug uptake. (3) Drugs delivered by these cells can be better localized in the tumor, reducing its interaction with normal cells and contributing to decreasing systemic toxicity. (4) Reduced amounts of drugs could be sufficient for treating patients. The use of MSCs as advanced therapy medicinal products (ATMPs) requires the optimization of the culture protocol that meets Good Manufacturing Practice (GMP) rules. Moreover, one of the most important aspects to take into consideration is the high number of cells required for their application in cell therapy areas; MSCs are present within adult human tissues at very low concentrations; therefore, in order to obtain a sufficient number of cells to guarantee patient treatment, adequate ex vivo cell expansion is mandatory.

For this purpose, we validated a protocol to prepare large-scale MSCs loaded with PTX (MSC-PTXs) using a bioreactor, obtaining a mean of 591.43 × 10^6^ MSC-PTXs from a single preparation process. These cells met all the GMP requirements to be used in clinical trials for the treatment of cancer patients, including those with TNBC [19].

Detectable PTX activity was evidenced at 2 h after MSC loading, reaching peak concentrations of 1.7–2.0 pg/cell at 144 h [18]. Indeed, in another study, we used PTX-primed MSCs against human pancreatic adenocarcinoma in transwell co-culture and showed a growth-inhibitory effect on CFPAC-1 cells [20]; moreover, paracrine effects of MSCs, acting without cell-to-cell contact, were demonstrated by Coccè et al. [21].

In this work, we focused on the activity of conditioned media released by PTX-primed MSCs. The aim of the present preclinical study is to explore the possibility of a cell therapy approach based on the use of MSCs loaded with PTX to treat TNBC patients. For this purpose, we evaluated the in vitro cell viability, migration and colony formation of two TNBC cell lines, namely, MDA-MB-231 and BT549, treated with MSC-PTX conditioned medium (MSC-CM PTX) in comparison with CM of MSCs not loaded with PTX (MSC-CM CTRL) and free PTX.

## 2. Results

### 2.1. Quantification of Paclitaxel in MSC-PTX Conditioned Medium

Untreated MSCs and MSCs loaded with PTX were cultured as described in [19,22]. The concentration of PTX in MSC-conditioned medium (MSC-CM) was evaluated after 24 and 48 h of incubation (PTX 24 h and PTX 48 h, respectively). We found a slight but non-significant difference between MSC-CM PTX 24 h and 48 h, which contained 222 ng/mL and 238 ng/mL of PTX, respectively. Nevertheless, although they carried the same amount of MSC-derived PTX, they may have had different cargos in terms of cytokines, small RNAs and other molecules, since they were collected at different times after MSC loading. Therefore, we performed experiments with both media. Conditioned media from MSCs incubated for 24 and 48 h with vehicle were used as controls (CTRL 24 h and CTRL 48 h, respectively). As expected, we detected no PTX in MSC-CM CTRL 24 h or 48 h.

### 2.2. Effects of MSC-CM PTX on the Viability of TNBC Cells

In the first set of experiments, TNBC cell lines (BT549 and MDA-MB-231) were incubated with the 24 h and 48 h CMs to evaluate whether the CMs had similar effects on both cell lines in a 144 h assay, as already reported in other cell models [18]. Having quantified the PTX contents in the MSC-CM PTX 24 h and 48 h, we were able to determine their activities in terms of PTX concentrations. The results obtained indicate that the effects of MSC-CM PTX (24 h and 48 h) were similar in both cell lines (Figure 1).

In fact, the survival inhibition was almost the same for both cell lines after 144 h of MSC-CM PTX 24 h treatment, with IC_50_ values equal to 0.99 and 1.00 ng/mL for MDA-MB-231 and BT549, respectively (Figure 1a). Similar results were obtained when the cells were incubated for 144 h (Figure 1b) with MSC-CM PTX 48 h. The IC50s obtained were analyzed, and there was no significant difference between the two treatments (i.e., CMs collected at 24 h and 48 h); moreover, they were similar for both cell lines, with no significant difference (Figure 1c). Since the inhibitory effect of MSC-CM PTX was substantial in both cell lines, in subsequent experiments we reduced the incubation of TNBC cells with CM to 72 h. The effects of MSC-CM PTX were then directly compared with the free-form drug. MDA-MB-231 and BT549 cells were cultured with increasing concentrations (1 pg/mL–100 ng/mL) of free PTX for 72 h (Figure 2a–c). As shown in Figure 2a, the average IC_50_ was 0.56 ng/mL for MDA-MB-231 and 0.93 ng/mL for BT549 treated with MSC-CM PTX 24 h, five-fold lower than that of free PTX (Figure 2c). Thereafter, we also tested MSC-CM PTX 48 h and obtained for MDA-MB-231 and BT549 IC_50_s of 0.50 and 1.6 ng/mL, respectively (Figure 2b). The average IC_50_ for free PTX was 5.2 ng/mL for MDA-MB-231 and 5.4 ng/mL for BT549 (Figure 2c). In the histogram reported in Figure 2d, the average IC50 values of MSC-CM PTX 24 h and 48 h obtained in both cell lines vs. free PTX are shown, with a *p*-value < 0.01. No significant difference was found between the 24 h and 48 h conditioned media.

As a further confirmation of these results, Trypan Blue cellular counts revealed that the number of total viable cells treated with MSC-CM-PTX 24 h was significantly lower than for MSC-CM-CTRL 24 h, as shown in Appendix A. Trypan blue counts serve as additional evidence that the cause of this viability loss was a cytotoxic effect, which was also validated by MTS assay. Furthermore, as shown in Appendix A, levels of Ki-67, a proliferation marker [23], were slightly decreased in cells treated with MSC-CM-PTX 24 h 1.11 ng/mL compared to the MSC-CM-CTRL 24 h at 48 h (*p* = 0.04 CM-PTX vs. CM-CTRL), but at 72 h no significant difference was measured, suggesting the absence of a mitotic block. This analysis could not be performed at a higher concentration, since only a few cells were detectable.

### 2.3. Effects of MSC-CM PTX on the Migration of TNBC Cell Lines

To confirm the inhibitory effects of MSC-CM PTX 24 h and 48 h on TNBC cell lines, we performed a wound healing assay (Figure 3).

Cells were able to achieve a significantly higher percentage of wound closure when treated with MSC-CM CTRL compared to MSC-CM PTX 1:200 dilution, corresponding to PTX concentrations of 1.1 ng/mL and 1.19 ng/mL for MSC-CM PTX 24 h and MSC-CM PTX 48 h, respectively. We chose these concentrations since they were similar to the IC_50_ values calculated after 72 h in viability assays in both cell lines. Representative images of MDA-MB-231 exposed to MSC-CM CTRL, MSC-CM PTX 24 h, MSC-CM PTX 48 h and free PTX 1 and 5 ng/mL are shown in Figure 3a. MDA-MB-231 cells treated with MSC-CM PTX 24 h or MSC-CM PTX 48 h migrated significantly less than cells exposed to MSC-CM CTRL or PTX at concentrations equal to 1 ng/mL (*p* < 0.001).

In Figure 3b, the results of the experiments performed on BT549 cells are reported. Representative images of the different treatments, taken at different times, are shown in the upper panels, and the quantifications of data obtained in three independent experiments are reported in the panels below. After 48 h of exposure to MSC-CM PTX 24 h or 48 h, migration was significantly inhibited compared to cells treated with CM-CTRL and free PTX at the same concentration (*p* < 0.01). Altogether, these data indicate that the migration of both TNBC cell lines was affected by conditioned medium of MSCs loaded with PTX. In particular, TNBC cell lines showed drastically reduced migration when treated with conditioned medium vs. free PTX at equal concentrations: possibly, MSC-CM PTX contained an additional agent(s) that increased efficacy.

### 2.4. Effects of MSC-CM PTX on TNBC Colony Formation

To confirm the tumorigenic potential of MDA-MB-231 and BT549 TNBC cells, we analyzed their clonogenic capabilities. We found that MDA-MB-231’s ability to form new colonies was blunted when treated with MSC-CM PTX 24 h or 48 h diluted 1:100 and 1:200, corresponding approximately to 2.22 ng/mL or 2.38 ng/mL and 1.11 ng/mL or 1.19 ng/mL of PTX, respectively (*p* = 0.001 vs. MSC-CM-CTRL), similar to free PTX 2 or 1 ng/mL (Figure 4a,b). Few colonies were detected in BT549 cells treated with the higher dilution. The experiments were performed in quadruplicate for each cell line. These data confirm that MSC-CM PTX concentrations were sufficient to achieve the same effects observed with 1 ng/mL or 2 ng/mL free PTX.

### 2.5. Caspases 3 and 7 Are the Effectors of Cell Viability Inhibition

As the conditioned media caused significant inhibition of viability, migration and colony formation in TNBC cell lines, we chose to test the activation of caspases 3 and 7. For this purpose, both cell lines were treated with DMEM; MSC-CM CTRL; MSC-CM PTX 24 h 1:50 and 1:100 (with CM-PTX concentrations equal to 4.44 ng/mL and 2.22 ng/mL, respectively); and PTX (5 ng/mL). These experiments were performed in parallel with viability assays, at least in triplicate.

As shown in Figure 5a, treatment of MDA-MB-231 with MSC-CM PTX 24 h significantly increased caspase 3/7 enzymatic activity after 48 h vs. the CTRL treatment (1:50 and 1:100 dilutions; *p* = 0.0001 and *p* = 0.0048, respectively). Similar effects were observed after 72 h.

These data were corroborated by cell viability data showing a significant reduction (MSC-CM PTX 1:50 vs. CTRL; *p* = 0.0207 at 48 h, *p* = 0.0003 at 72 h; Figure 5b). Using the conditioned medium diluted 1:100, viability decreased after 72 h, with *p* = 0.0001, vs. CTRL.

The data were further confirmed by immunoblotting at 72 h showing enhanced levels of cleaved Caspase 3 in cells treated with MSC-CM PTX vs. CTRL. The experiments were performed in triplicate. We show a representative image of immunoblotting (Figure 5c) and the quantification of Cleaved-Caspase 3 normalized vs. GAPDH (Figure 5d; MSC-CM PTX 1:50 or 1:100 vs. CTRL; *p* = 0.0012 and 0.0033, respectively).

In the BT549 cell line, Caspase 3/7 activity was significantly increased both at 48 and 72 h with MSC-CM PTX 1:50 (*p* < 0.0001 and *p* = 0.0028 vs. CTRL, respectively) (Figure 5e), while cell viability was decreased at 48 h with MSC-CM PTX 1:50 vs. CTRL treatment (*p* = 0.027), and the effect was more evident after 72 h (*p*-value = 0.0055) (Figure 5f).

All these data were also confirmed by immunoblotting at 72 h with enhanced levels of cleaved Caspase 3 normalized vs. GAPDH of cells treated with MSC-CM PTX 1:50 and 1:100 vs. CTRL, respectively, *p* = 0.0002 and *p* = 0.0022. The experiments were performed in triplicate (Figure 5g,h).

## 3. Discussion

Despite several advances in the treatment of TNBC, it remains a highly aggressive disease, with a significantly shorter overall survival compared with other breast cancer subtypes, reaching a 5-year mortality of about 40% [24]. Neoadjuvant chemotherapy remains the standard of care for most stage II–III TNBC patients [25]. PTX is one of the recommended options of treatment [26] in both (neo-)adjuvant and metastatic settings. Peripheral neurotoxicity of PTX leads to treatment discontinuation in approximately 20% of patients [27]. Our research tested an innovative strategy to optimize PTX therapy with the aim of reducing specific toxicities while maintaining similar activity levels. In fact, several studies have suggested that the development of TIPN could be associated with functional decline, increased risk of falls and diminished quality of life [28].

The incidence and severity of TIPN varies according to different factors: several studies have reported that the cumulative incidence and prevalence of TIPN symptoms in women with breast cancer receiving taxane-based chemotherapy ranges from 57 to 83%, with 2 to 33% developing severe neuropathy [29].

The use of MSCs primed with PTX has been suggested as a novel approach to overcome toxicity issues. In principle, PTX-loaded MSCs may exert anti-cancer effects both by direct cell–cell contact and by paracrine action of the released drug [21]. We previously estimated the kinetics of PTX release and retention by MSCs [18]. In this work, we focused on the activity of conditioned media released by PTX-primed MSCs. The concentrations of PTX quantified in the CM after 24 h and 48 h of incubation were nearly equivalent. In fact, treatments using MSC-CM PTX 24 h and 48 h inhibited TNBC cell line viabilities at 72 h and 144 h in a similar way. Interestingly, the IC50s after 3 days were five-fold lower than free PTX in MDA-MB-231 and 2.5-fold lower in BT549, suggesting that the use of CM could allow dose reduction and lead to lower toxicity in patients. Indeed, higher inhibition of migration was obtained by MSC-CM PTX compared to free PTX at the same concentration. We found that MSC-CM PTX with a PTX concentration of 1 ng/mL inhibited TNBC cell migration, and we also demonstrated that conditioned media with PTX concentrations of 1 and 2 ng/mL had the same effects as 1 and 2 ng/mL free PTX in terms of inhibiting tumorigenicity and dissemination capability in colony formation assays, as validated in other cancers [8].

As reported by Scioli et al. [14], who found a significant cytotoxic effect of adipose-derived MSCs loaded with PTX in human breast cancer cell lines, we demonstrated with enzymatic assays that MSC-CM PTX-induced cell apoptosis after 48 and 72 h was directly proportional to cleaved Caspase 3 protein levels in both human TNBC cell lines. Apoptosis activation has been also confirmed in other cancer cell lines, such as malignant pleural mesothelioma [21].

According to our previous experience, we could speculate that the increased efficacy of MSC-CM PTX compared with free PTX may be due to several factors: encapsulation of PTX in extracellular vesicles [16,17,30,31], cytokine or chemokine release by MSCs, or the presence in the MSC-CM of additional compounds, such as miRNAs or lncRNAs.

Breast cancer is the most frequent cancer in women worldwide, and it is increasing particularly in developing countries. Neoadjuvant therapy has been demonstrated to improve EFS only when pCR has been reached. It is therefore very important to maintain the dose intensities of drugs, which are often reduced or in some cases withdrawn due to related toxicity. The aim of our study was to evaluate the in vitro efficacy of MSC-CM PTX vs. free PTX in TNBC cell lines. Our results confirm that MSC-CM PTX has anti-proliferative, anti-tumorigenic and pro-apoptotic effects stronger than those of free PTX. Based on in vitro preclinical data, we plan to evaluate the efficacy and the safety of this delivery method in murine xenografts and syngeneic models. Recently, Mocchi and colleagues demonstrated a method that combines ultrafiltration and freeze-drying to transform MSC-CM into a pharmaceutical product [32], making it feasible to test in vivo the use of CM for treating TNBC. Our final goal is to improve the tolerability and efficacy of treatments by optimizing their delivery. TIPN is a very disabling condition, which can last up to 12 months after the completion of treatment, also causing social and working impairment. By identifying new methods to deliver drugs such as PTX, we can set up a model applicable to other anti-cancer drugs, thus moving towards a personalized approach to patients and cancer.

## 4. Materials and Methods

### 4.1. Cancer Cells and Compounds

MDA-MB-231 and BT549 cells were purchased from the American Type Culture Collection (ATCC). MDA-MB-231 and BT549 cells were maintained in Dulbecco’s modified Eagles medium high glucose supplemented with 10% Fetal Bovine Serum (FBS), 1 mM L-glutamine and 100 U/mL penicillin-streptomycin and were grown in a humidified incubator at 37 °C and 5% CO_2_. Paclitaxel from Taxus Brevifolia (T7402) (PTX) was purchased from Merck Life Science (Milano, Italy).

### 4.2. Adipose Tissue Collection

Adipose tissue (AT) lipoaspirates were collected, under general anesthesia, from healthy volunteer donors (age range: 18–66 years) undergoing plastic surgery for aesthetic purposes. Samples were collected after signed informed consent was provided with no objection to the use of surgical tissues for research (otherwise eliminated), in accordance with the Declaration of Helsinki. The informed consents were obtained prior to tissue collection. The Ethics Committee of Regione Lombardia, the Institutional Review Board Section of the IRCCS Neurological Institute C. Besta Foundation, approved the design of the study (verbal number 29, 4 May 2016) reported by Lisini et al. in 2020 [19]. Samples were processed within 24 h from surgery.

### 4.3. MSC Isolation, Large-Scale Expansion and Loading with PTX

MSCs from AT lipoaspirates were isolated according to the GMP rules for ATMP production processes, as previously described [19,22].

Briefly, after disaggregation by enzymatic digestion with 200 U/mL of collagenase type I Life Technologies, Carlsbad, CA, USA), the samples were centrifuged (1000× *g*, 15 min) and the floating fractions and cellular pellets were plated on 150 cm^2^ flasks (Euroclone, Milano, Italy), 10 mL/flask, and cultured in DMEM (Euroclone, Milano, Italy) supplemented with 5% platelet lysate Stemulate (Cook Reagent, Indianapolis, IN, USA) and 2 mM L-glutamine (Euroclone, Milano, Italy) until at least 20 × 10^6^ cells were obtained at a passage not exceeding P3.

MSC large-scale expansion was carried out using a bioreactor, the Quantum Cell Expansion System (Terumo BCT, Lakewood, CO, USA). After uploading the disposable expansion set (Terumo BCT), the bioreactor was coated overnight with 5 mg of human fibronectin (Corning Incorporated, Deeside, UK) to promote cell adhesion; then, at least 20 × 10^6^ MSCs, expanded as described above, were seeded in the system at an inlet rate of 25 mL/min, followed by a 24 h phase in which MSCs recirculated and adhered to the support. Since cells are not visible in the hollow fiber, cell growth was estimated according to lactate generation by the cells in the system. Fresh complete medium was added continuously to cells, and the inlet rate was adjusted as required by the rate of lactate generation. Drug loading of MSCs with PTX was started at the end of the expansion phase, when the lactate value increased to less than 0.5 mM in 24 h, and occurred directly in the bioreactor. A 4 L complete medium bag, supplemented with PTX at a final concentration of 10 µg/mL (PTX, TEVA, Milano, Italy, 6 mg/mL), was prepared and connected to the instrument.

Through the “touch screen”, the medium in the instrument was completely replaced with medium supplemented with PTX, within 5 min. This phase was followed by the recirculation of the medium in the Quantum disposable set for 24 h at an inlet rate of 0.1 mL/min. After 24 h, MSCs loaded with PTX were washed with PBS (Euroclone), detached from the expansion set using recombinant trypsin (TrypLE Select 1X, Thermofisher, Waltham, MA, USA) and eluted in a 500 mL bag using complete medium.

Both MSC primary cultures and expanded MSC-PTXs were analyzed for number and viability, population doubling time (PDT), expression of the typical MSC markers (CD90, CD73 and CD105) and multi-differentiation ability towards mesodermal lineages (osteogenic, adipogenic and chondrogenic differentiation), as previously described [19,22,33].

### 4.4. Preparation of Conditioned Media

Conditioned media from MSC-PTXs (MSC-CM PTX) were prepared by seeding MSC-PTXs in 48-multiwell plates (24,000 cells/well, 350 μL/well); the cells were seeded in separate wells for the collection of conditioned media (CMs) at the two time points after seeding, i.e., 24 h and 48 h. The plates were maintained at 37 °C, 5% CO_2_, and CMs were collected at the desired time. Hence, the two MSC-CM PTXs (24 h and 48 h) represent two different preparations as a consequence of the different release times of the chemotherapeutic drug (PTX) from the MSCs. Vehicle-treated MSCs were cultured to obtain control conditioned media (MSC-CM CTRL).

### 4.5. Paclitaxel Dosages

Quantification of PTX was performed using an LC/MS 6546 platform which included an Agilent 1290 II LC system (Agilent Technologies, Palo Alto, CA, USA) coupled to a time-of flight mass spectrometer (Agilent Technologies). Chromatographic separation was achieved using a Zorbax Eclipse Plus C18 Column (50 × 2.1 mm, 1.8 μm) (Agilent Technologies). Mobile Phase A was 100% water, and phase B was methanol, both containing 0.2% of formic acid. The following elution gradient was used: 0 min 20% of A, 3 min 80% of A, 4 min 80% of A, 4.1 min 20% of A, 5 min 20% of A. The flow rate was 0.5 mL/min, the column temperature was set at 45 °C and the injection volume was 5 μL. Samples were analyzed in ESI positive ionization mode, and the mass spectrometer operated at a resolving power of 40,000 over a full scan range of m/z 50–1100 at a scan rate of 2 spectra/s with the following settings: gas flow 10 L/min; gas temperature 180 °C; nebulizer 50 psi; sheath gas temperature 350 °C; sheath gas flow 11 L/min; capillary voltage 3500 V; nozzle voltage 1000 V; and fragmentator at 175 V. Purine was used as reference mass (m/z = 121.0509) and continuously infused at a flow rate of 0.08 mL/min. Quantification and detection of PTX was performed using the Na adduct ion, and a linear range from 0.1 ng/mL to 500 ng/mL was used for quantification. Samples were processed as follows: 150 μL of cold methanol was added to 50 μL of sample and vortexed. Then, samples were centrifuged for 15 min at 15,000× *g*, and supernatants were analyzed by LC-MS.

### 4.6. Viability Assay

Cells were seeded into 96-well plates (5000 cells/well or 3000 cells/well for reading after 72 h or 144 h, respectively) and incubated for 24 h. Growth medium was then replaced with conditioned medium (MSC-CM CTRL and MSC-CM PTX 24 h/48 h) at progressively lower sequential concentrations. The CMs were differently diluted for 72 h and 144 h because at this later time the CMs showed higher toxicity. Growth inhibition was quantified after 72 h and 144 h using CellTiter-AQueous (Promega, Madison, WI, USA), according to the manufacturer’s instructions. Relative cell viability percentage was obtained by first subtracting the blank with no cells (CellTiter-Aequos reagent in medium), then normalizing vs. the MSC-CM CTRL. The absorbance was measured using a microplate reader (TECAN) with an excitation and emission of 490 nm. IC_50_ was determined using a sigmoidal regression model using GraphPad Prism 6.0 (GraphPad Software, La Jolla, CA, USA) and was defined as the PTX concentration in CM required to inhibit growth by 50%. Each experiment was repeated at least two or three times with four or five replicates for both cell lines.

### 4.7. Wound Healing Assay

TNBC cell line cells were seeded 10^6^/well in a 6-well plate and grown until they reached 90% confluence. A scratch was made in the cell monolayer with a sterile pipette tip, and cells were cultured in reduced FBS (2%) medium under the following conditions: MSC-CM CTRL, MSC-CM PTX 24 h and MSC-CM PTX 48 h diluted 1:200 or in the presence of 1 ng/mL PTX. A low serum concentration was used to suppress cell proliferation that may have confounded the migration results. Images were taken at 0, 24 and 48 h to measure the migration capacity of the cells. The scratch area was calculated using ImageJ software, https://imagej.nih.gov/ij/ [34] and each condition was normalized vs. its own T0 to obtain the percentage of wound closure. Each experiment was repeated in triplicate for both cell lines.

### 4.8. Clonogenic Survival Assay

MDA-MB-231 cells were seeded at 300 cells/well in 24-well plates and at 1000 cells/wells in 6-well plates, while BT549 cells were seeded at 450 cells/well in 12-well plates and at 1500 cells/well in 6-well plates. After 24 h, the medium was replaced with MSC-CM CTRL and MSC-CM PTX 24 h and 48 h diluted 1:100 and 1:200 or medium containing 1 and 2 ng/mL of PTX. After 14 days for MDA-MB-231 and 21 days for BT549, the cells were washed with PBS, fixed with methanol for 10 min at room temperature, then washed twice with PBS. Hence, the cells were stained with 0.5% crystal violet, 25% methanol, and images were captured with a Biorad Instrument camera (Bio-Rad, Hercules, CA, USA). Analysis was performed using ImageJ software. We calculated the number of colonies for each experiment, and we presented the clonogenic data as % of colony formation units (% CFUs). Each experiment was repeated four times for both cell lines.

### 4.9. Caspase Activity Assay and Viability Assay

Measurements of caspase activities in cells were performed using the commercially available Caspase-Glo 3/7 Assay (Promega, Madison, WI, USA), according to the manufacturer’s instructions. Luminescence was measured using a microplate reader (Tecan Trading AG, Switzerland) at 48 and 72 h. Both cell lines were cultured in DMEM, MSC-CM CTRL, MSC-CM PTX 24 h 1:50 and 1:100 and PTX 5 ng/mL. At the same time points, measurements of cell viability were performed using the commercially available CellTiter-Glo Assay (Promega, Madison, WI, USA), which was mixed at a 1:1 ratio with supernatant from the treatment plate. The mix was incubated for 3 min at room temperature on the shaker and then incubated for 11 min at room temperature, followed by luminescence measurement. The caspase activity associated with viability assay was performed in three independent experiments.

### 4.10. Immunoblotting Analysis

Cell lysates from parental and resistant cells were prepared in RIPA buffer with protease and phosphatase inhibitors, and 40 μg of total proteins were loaded and run on SDS-PAGE with specific antibodies: monoclonal mouse anti-human GAPDH (Merk Life Science S.r.l. Milano, Italy, 1:2000 dilution) and monoclonal mouse anti-human c-Caspase 3 (Cleaved Caspase 3 (Asp175) (5A1E) #9664, Cell Signalling, 1:1000 dilution). The proteins were blotted on nitrocellulose membranes and blocked in 5% non-fat milk. The incubation of primary antibodies was run overnight at 4 °C. The secondary antibodies (Bio-Rad, anti-mouse, 1:2000 dilution, and anti-rabbit 1:2000 dilution) were incubated for 1 h at room temperature. Chemiluminescence signals were detected using the ChemiDoc XRS + Imaging System (Bio-Rad, Hercules, CA, USA). This experiment was performed in triplicate.

### 4.11. Statistical Analysis

Statistical analyses were performed using GraphPad Prism 6 Software. Quantitative data, collected from independent experiments, were expressed as the means ± standard deviations (SDs). Differences between two data sets were determined by two-sided *t*-tests or unpaired *t*-tests with Welch’s correction. Wound healing results were analyzed using two-way ANOVA and Bonferroni tests for multiple comparisons. A *p*-value lower than 0.05 was considered significant.

## Figures and Tables

**Figure 1 ijms-24-05864-f001:**
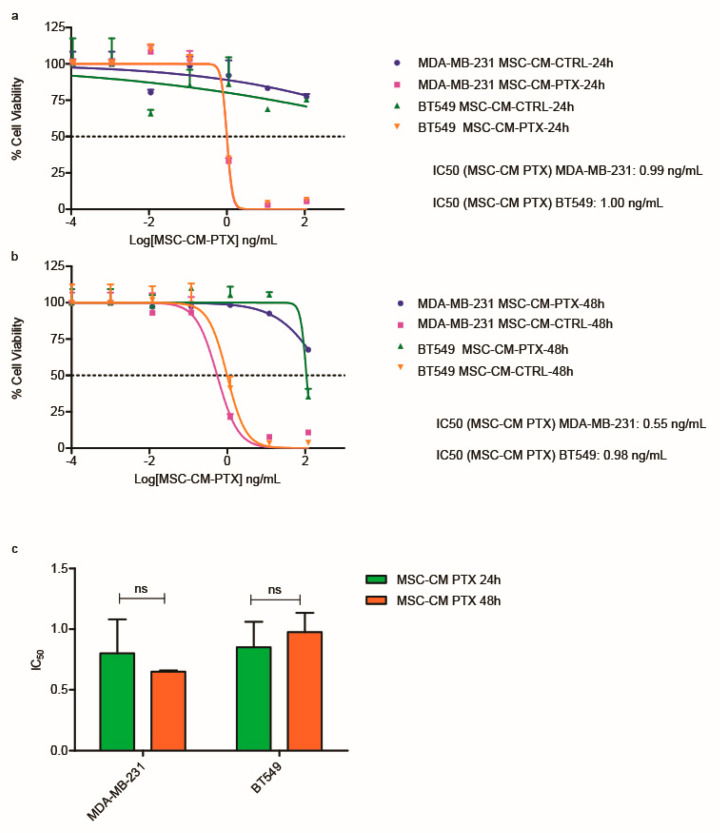
Viability inhibition in TNBC cell lines using MSC-CM PTX 24 h and 48 h at 144 h. Effects of MSC-CM PTX 24 h and 48 h at 144 h on the survival of TNBC cell lines (**a**–**c**). Dose–response curves in MDA-MB-231 and BT549 cells showing the effects of treatment with serial concentrations of MSC-CM PTX 24 h (**a**) and MSC-CM PTX 48 h (**b**). Dashed lines depict IC50 values. Both cell lines were tested in triplicate with both treatments and controls. A representative curve is shown for both experiments. The histogram reports the mean IC50 values for both cell lines treated with MSC-CM PTX 24 h and 48 h in triplicate (**c**). The data are expressed as means ± standard deviations (SDs). Differences between two data sets were determined by unpaired *t*-tests.

**Figure 2 ijms-24-05864-f002:**
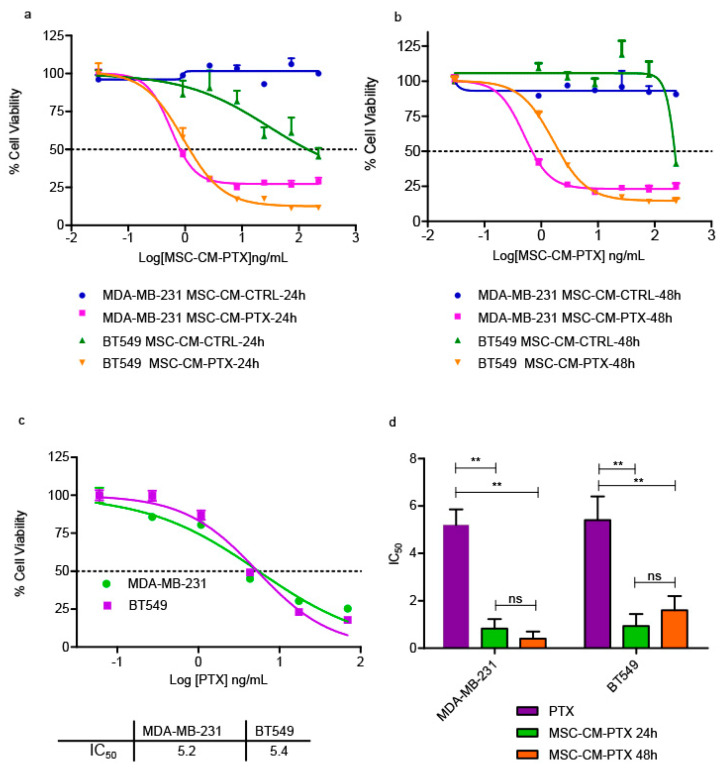
Viability inhibition in TNBC cell lines treated with MSC-CM at 72 h. Dose–response curves in MDA-MB-231 and BT549 cells showing the effects of treatment with different concentrations of MSC-CM PTX 24 h (**a**), 48 h (**b**) and free PTX (**c**). Dashed lines depict IC50 values. Both cell lines were examined in two independent experiments. Points, mean of duplicate (CM treatments) and triplicate (free PTX treatments) determinations, and error bars described SEs. The histogram reports the mean IC50 values ± the standard deviations (SDs) for both cell lines treated with MSC-CM PTX 24 h and 48 h in duplicate (**d**). Unpaired *t*-tests determined the *p*-values: ** *p* < 0.01.

**Figure 3 ijms-24-05864-f003:**
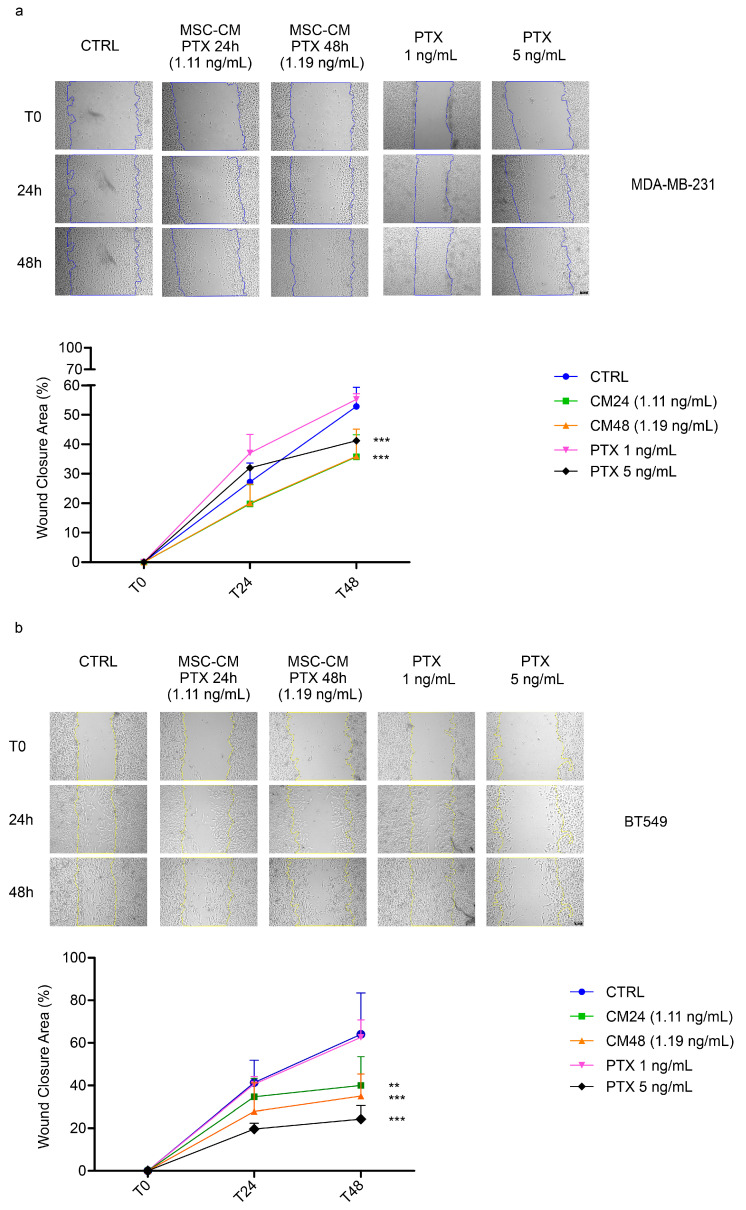
MSC-CM PTX inhibits migration in TNBC cell lines. Scratch wound healing assays in both cell lines, MDA-MB-231 (**a**) and BT549 (**b**), exposed to MSC-CM CTRL, CM PTX 24 h, CM PTX 48 h and free PTX (1 ng/mL and 5 ng/mL). Images were captured at time 0 and after 24 h and 48 h. For both cell lines, representative images at T0-24 h-48 h with the indicated treatments are shown. This experiment was performed in triplicate. The graphs show quantifications of wound closure over time from three experiments ((**a**,**b**), bottom). The scratch area was calculated using ImageJ software, and each condition was normalized with respect to its own T0 to obtain the percentage of wound closure area. For the analysis of the data, two-way ANOVA was used with Bonferroni correction to determine *p*-values: ** *p* < 0.01, *** *p* < 0.001; error bars indicate SDs. Scale bar: 100 µm.

**Figure 4 ijms-24-05864-f004:**
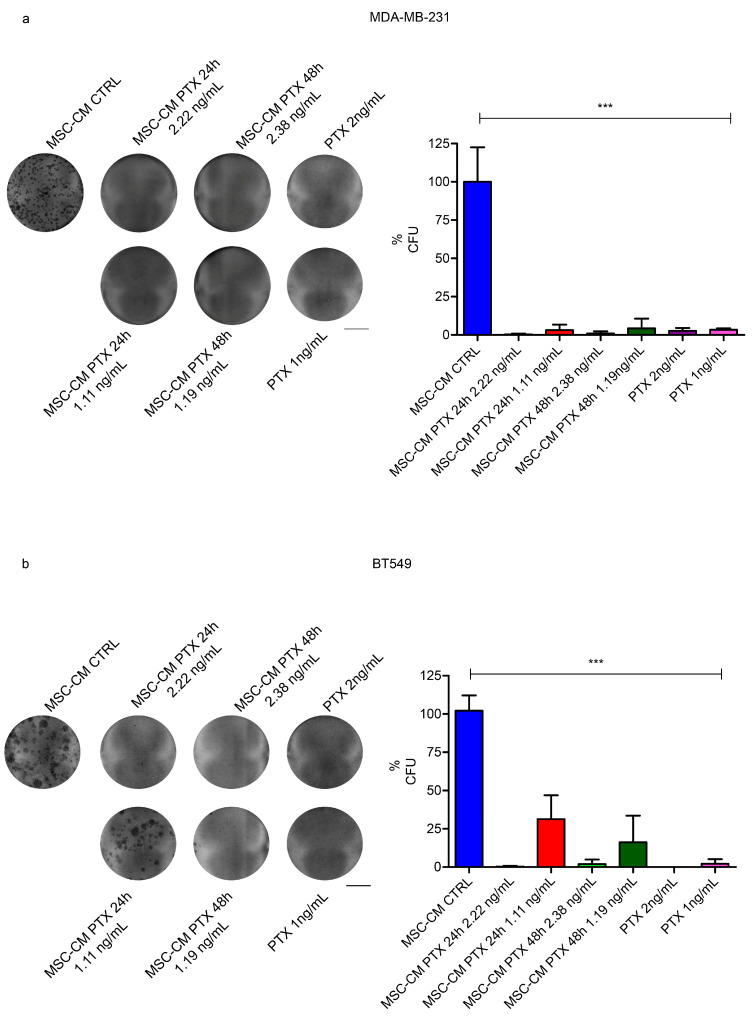
MSC-CM PTX 24 h/48 h effects on TNBC cell line colony formation. MDA-MB-231 and BT549 cells were seeded at very low densities, and single-cell colonies were counted after 14–21 days. Representative images of clonogenic assays (left) and histograms reporting the percentages of colony formation units (%CFUs) (right) of MDA-MB-231 (**a**) and BT549 (**b**) under different conditions are shown. Scale bar is equal to 10 mm. The experiment was performed in quadruplicate; unpaired *t*-tests determined *p*-values: *** *p* < 0.001; error bars indicate SDs. Scale bar: 10 mm.

**Figure 5 ijms-24-05864-f005:**
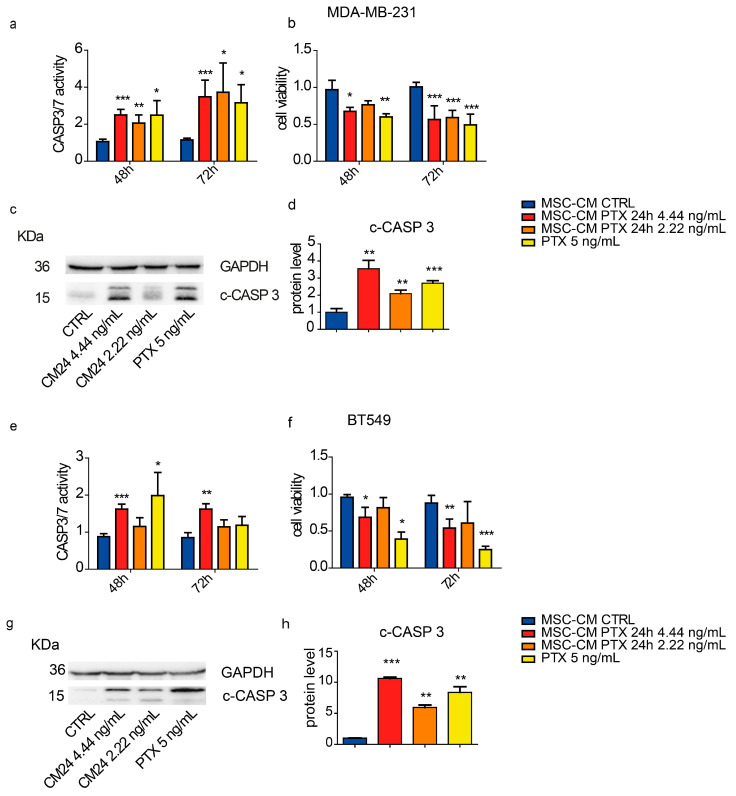
Effect of MSC-CM PTX on Caspase 3/7 activity and cell viability of TNBC cells. Caspase 3/7 activity (**a**,**e**) and cell viability (**b**,**f**) of TNBC cells were measured after treatment with 1:50 (4.44 ng/mL) and 1:100 (2.22 ng/mL) MSC-CM PTX 24 h dilution or with free PTX. MDA-MB-231 and BT549 were seeded 5000 cells/well in 96-well plates and exposed to two dilutions (1:50 and 1:100) of MSC-CM PTX 24 h for 48 and 72 h. Caspase-3/7 activity was measured using the luminescent Caspase-Glo^®^ 3/7 Assay, and the results were normalized vs. MSC-CM CTRL, expressed as caspase activity fold changes over CTRL (**a**,**e**). On the right, the viability assay analysis is shown (**b**–**f**). All data points represent the means of the assays performed in triplicate ± SDs (n = 3). Representative images of Western blotting (**c**–**g**) and quantification (**d**–**h**) of three independent experiments showing cleaved caspase 3 protein expression. Error bars describe SDs, and unpaired *t*-tests determined *p*-values: * *p* < 0.05, ** *p* < 0.01, *** *p* < 0.001.

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
