# Peer review of "Conditioned Medium of Mesenchymal Stromal Cells Loaded with Paclitaxel Is Effective in Preclinical Models of Triple-Negative Breast Cancer (TNBC)"

_ijms, 2023, doi:10.3390/ijms24065864_

Round 1

Reviewer 1 Report

Review about Conditioned Medium of Mesenchymal Stromal Cells loaded with Paclitaxel is effective in preclinical models of Triple-Negative Breast Cancer (TNBC).

 The authors' aim with the present preclinical study is to explore the possibility of a cell therapy approach based on the use of MSC loaded with paclitaxel (PTX) to treat triple negative breast cancer (TNBC) affected-patients.

The main flaw of the article that the authors could not prove that PTX has any effect on the TNBC cells, at least the provided data do not support this assumption, however the media of the PTX loaded MSC cells had effect.

If I understood well, the Mesenchymal Stromal Cells (MSC) were loaded with PTX and the conditioned medium (MSC-CM PTX) was used for treating the cell lines of MDA-MB-231 and BT549. The conditioned media (24 h/48 h) of the cell lines contained different amount of PTX. The difference between the 24h and 48h incubation of PTX 222/ 238 ng/mL is negligible, there was no necessary to do every experiment with the two media.

In the Fig1. I cannot understand, why the viability of the control cells did not start from 100 %. Usually the viability tests are based on redox reactions. I would suggest to count the cells simply before / after treatments. Further, the control should not have scattering if yes, why?.

Further I have problem with curve fit in every Figs, mostly in the Fig2. The plotted curves of MDA-MB-231 MSC-CM PTX 24h/ 48h do not related to the measured data, at all. The curves seem to be random plots, although the measurements seemed to be correct.

Mostly the migration assay (scratch wound assay) corroborated that PTX did not have effect, at all, but the MSC-CM PTX had.

In summary, I suggest to use  2 ng/mL PTX as positive control in every experiment. I drop the 48h media, because there is no significant difference to the 24h samples. The provided data did not support that PTX would have effect, however the  MSC-CM PTX media should contain some agent(s) that had real efficacy. Please use better program for fitting the curves for the measured values.

Author Response

Monza, 09 February 2023

Dear Reviewers,

please find enclosed a manuscript entitled " Conditioned Medium of Mesenchymal Stromal Cells loaded with Paclitaxel is effective in preclinical models of Triple-Negative Breast Cancer (TNBC)", which I am submitting for publication in IJMS. The manuscript describes in vitro the efficacy of MSC-CM PTX vs free PTX on TNBC cell lines.

The manuscript has been reviewed by all and they have approved the manuscript for submission to IJMS. The results shown in this manuscript are not published or under consideration for publication by any other journal.

We declare no conflict of interest.

Best regards,

Nicoletta Cordani, PhD

School of Medicine and Surgery

University of Milan-Bicocca

Reviewer 2 Report

REVIEW

This is a critique of the manuscript from Cordani et al titled “Conditioned Medium of Mesenchymal Stromal Cells loaded with Paclitaxel is effective in preclinical models of Triple-Negative Breast Cancer (TNBC)” submitted to the International Journal of Molecular Sciences.

Major Issues:

The English requires extensive editing.

The overall hypothesis is that MSCs loaded with Paclitaxel will hone to TNBC tumors and deliver the drug to the cancer cells leading to the destruction of the tumor. However, loaded MSCs are not used in the experiments and data presented only conditioned media. Thus, the data presented do specifically address the overall stated goal. The data would be stronger while also answering the question “how long do the MSC remain loaded with Paclitaxel?” if the loaded MSCs were co-cultured with the TNBC. This could be accomplished through traditional 2D culture systems or by use of Transwell inserts where the loaded MSCs were cultured suspended above the TNBC.

Minor issues:

Line 44- HER2 is Human Epidermal Growth Factor Receptor-2

Figure 3 needs scale bars or magnification indication in the legend. The overlapping error bars between CTRL and MSC-CM PTX (1.11) in Panel B suggest there is not a significant difference even though it is stated as so in the text.

Figure 4 needs scale bars or magnification indication in the legend.

The font style and size used in the figures is hard to read especially Fig 5d.

Author Response

(The authors gave the same response as above.)

Round 2

Reviewer 1 Report

Reviewer comment on the answer of the ijms-2200007- v2.

The authors claimed that the MTS assay is much more sensitive and consistent than the operator cell counts. No, it is not true. Unfortunately I cannot agree with this. MTS/ MTT/ XTT assays based on redox reactions that can be effected by several things such as free radicals, not just the reductive capacity of the cells. I am aware of that it is used to show the healthiness of cells, but they should be applied with controls. That is why the untreated control cells had less values than 100%. I would like to see a control with PTX and MTS assay, whether there is no cross reaction.

Please use KI67 staining to show the decreased level of proliferation (Li et al., 2014; doi:10.3892/mmr.2014.2914).

I know what the wound healing assay for. Why do you expect reduced motility upon administration of PTX. Obviously dead cells do not migrate, however PTX and other toxins can have effect on migration; they can accelerate the movement sometimes.

Author Response

Response to Reviewer 1 Comments

The authors claimed that the MTS assay is much more sensitive and consistent than the operator cell counts. No, it is not true. Unfortunately I cannot agree with this. MTS/ MTT/ XTT assays based on redox reactions that can be effected by several things such as free radicals, not just the reductive capacity of the cells. I am aware of that it is used to show the healthiness of cells, but they should be applied with controls. That is why the untreated control cells had less values than 100%. I would like to see a control with PTX and MTS assay, whether there is no cross reaction.

Author response: Thank you for your comment, we performed cell count and in parallel MTS viability assay, before and after treatment, as shown in Supplementary Figure 1; these data corroborated previous results. As a technical control of MTS, we verified that the absorbance of CM-PTX and CM-CTRL with MTS solution did not show cross-reaction.

Please use KI67 staining to show the decreased level of proliferation (Li et al., 2014; doi:10.3892/mmr.2014.2914).

Author response: We thank the Reviewer for this suggestion, which helped clarify an important issue. As per your advice, we stained with anti-Ki67 the cells treated with CM-PTX 1.11 ng/mL, or CM-CTRL, or with DMEM, as reported in Supplementary Figure 2. Unpaired t-test analysis showed mild inhibition after 48h but not at 72h treatment, indicating that there is no clear evidence of proliferation inhibition; rather, these data point to a cytotoxic effect, also proven by apoptosis assays, as shown in figure 5.

I know what the wound healing assay for. Why do you expect reduced motility upon administration of PTX. Obviously dead cells do not migrate, however PTX and other toxins can have effect on migration; they can accelerate the movement sometimes.

Author response: We agree with the Reviewer’s comment; however, we were requested to include PTX as a positive CTRL in migration assays. We tried with 1 and 2 ng/mL and did not see any effect, so we tried with 5 ng/mL. By contrast, MSC-CM PTX suppressed migration already at 1 ng/mL. It is possible, as it was commented by Reviewers in the first round of revision, that “the  MSC-CM PTX media should contain some agent(s) that had real efficacy”. We added this comment to the manuscript.

Reviewer 2 Report

My previous critique was "The overall hypothesis is that MSCs loaded with Paclitaxel will hone to TNBC tumors and deliver the drug to the cancer cells leading to the destruction of the tumor. However, loaded MSCs are not used in the experiments and data presented only conditioned media. Thus, the data presented do specifically address the overall stated goal." The authors have addressed this in their response but not sufficiently in the manuscript. I suggest adding words or a sentence to the last paragraph of the Introduction stating what was written in the Response.

Author Response

Response to Reviewer 2 Comments

My previous critique was "The overall hypothesis is that MSCs loaded with Paclitaxel will hone to TNBC tumors and deliver the drug to the cancer cells leading to the destruction of the tumor. However, loaded MSCs are not used in the experiments and data presented only conditioned media. Thus, the data presented do specifically address the overall stated goal." The authors have addressed this in their response but not sufficiently in the manuscript. I suggest adding words or a sentence to the last paragraph of the Introduction stating what was written in the Response.

Author response: We thank the reviewer; following his/her advice, we added a sentence to the last paragraph of the Introduction.

Round 3

Reviewer 1 Report

In the Ki-67 staying was any detergent used such as Triton-X100, or Twin20, because it was not mentioned?

Author Response

Response to Reviewer 1 Comments

In the Ki-67 staying was any detergent used such as Triton-X100, or Twin20, because it was not mentioned?

Author response: Thank you for your comment, I forgot to insert Triton-X100 in the recipe of GDB buffer: 0.02M sodium phosphate buffer, pH 7.4, containing 0.45M NaCl, 0.2% (w/v) bovine gelatine, 0.2% Triton-X100.

It has been inserted in the text.
